# A Single Domain Shark Antibody Targeting the Transferrin Receptor 1 Delivers a TrkB Agonist Antibody to the Brain and Provides Full Neuroprotection in a Mouse Model of Parkinson’s Disease

**DOI:** 10.3390/pharmaceutics14071335

**Published:** 2022-06-24

**Authors:** Emily Clarke, Pawel Stocki, Elizabeth H. Sinclair, Aziz Gauhar, Edward J. R. Fletcher, Alicja Krawczun-Rygmaczewska, Susan Duty, Frank S. Walsh, Patrick Doherty, Julia Lynn Rutkowski

**Affiliations:** 1King’s College London, Institute of Psychiatry, Psychology and Neuroscience, Wolfson Centre for Age-Related Disease, Guy’s Campus, London SE1 1UL, UK; emily.clarke@kcl.ac.uk (E.C.); edward.fletcher@ucl.ac.uk (E.J.R.F.); alicja.krawczun-rygmaczewska@kcl.ac.uk (A.K.-R.); susan.duty@kcl.ac.uk (S.D.); patrick.doherty@kcl.ac.uk (P.D.); 2Ossianix, Inc., Gunnels Wood Rd., Stevenage SG1 2FX, UK; pawel@ossianix.com (P.S.); liz@ossianix.com (E.H.S.); aziz@ossianix.com (A.G.); walsh@ossianix.com (F.S.W.)

**Keywords:** TrkB, agonist antibody, variable new antigen receptor (VNAR), neuroprotection, transferrin receptor 1 (TfR1), blood-brain barrier (BBB), 6-OHDA, Parkinson’s disease

## Abstract

Single domain shark antibodies that bind to the transferrin receptor 1 (TfR1) on brain endothelial cells have been used to shuttle antibodies and other cargos across the blood brain barrier (BBB) to the brain. For these studies the TXB4 brain shuttle was fused to a TrkB neurotrophin receptor agonist antibody. The TXB4-TrkB fusion retained potent agonist activity at its cognate receptor and after systemic administration showed a 12-fold increase in brain levels over the unmodified antibody. Only the TXB4-TrkB antibody fusion was detected within the brain and localized to TrkB positive cells in the cortex and tyrosine hydroxylase (TH) positive dopaminergic neurons in the substantia nigra pars compacta (SNc), where it was associated with activated ERK1/2 signaling. When tested in the 6-hydroxydopamine (6-OHDA) mouse model of Parkinson’s disease (PD), TXB4-TrkB, but not the unmodified antibody, completely prevented the 6-OHDA induced death of TH positive neurons in the SNc. In conclusion, the fusion of the TXB4 brain shuttle allows a TrkB agonist antibody to reach neuroprotective concentrations in the brain parenchyma following systemic administration.

## 1. Introduction

The interaction of neurotrophins (NGF, BDNF, NT3 and NT4) with their cognate Trk receptors (TrkA, TrkB and TrkC, respectively) protects neurons from naturally occurring cell death during development [1,2]. Their ability to nurture developing neurons spawned numerous studies to determine if they can promote the survival of adult neurons, particularly in the context of neurodegenerative disease or acute brain injury [3,4]. In this context, promising results have been found with BDNF which, by activating the TrkB receptor, can protect neurons from death in, for example, preclinical models of PD [5], Alzheimer’s disease (AD) [6,7], and ischemic lesions [8,9,10,11]. In addition, BDNF can promote functional recovery of injured neurons following spinal cord injury [12,13,14] and stimulate the production of new neurons in the adult brain [15,16]. The loss of BDNF has also been suggested as a contributory factor to the progression of PD [17,18,19], AD [20] and Huntington’s disease [21,22,23], as well as to conditions such as depression [24,25].

However, the therapeutic potential of BDNF in neurodegenerative diseases, acute brain injury and other neurological conditions has not been realized in the clinical setting due in part to a short plasma half-life in vivo [26], exclusion from the brain parenchyma following systemic administration, and poor diffusion throughout the parenchyma due to a high isoelectric point [27]. Agonist antibodies that directly bind the TrkB receptor and mimic the neurotrophic activity of BDNF provides a long in vivo half-life, but the challenge of poor blood brain barrier (BBB) penetration remains. This has generally limited the systemic delivery of TrkB antibody to peripheral disorders such as obesity [28,29] and peripheral neuropathy [30]. Nonetheless, when delivered directly across the BBB by intracerebroventricular injection prior to an ischemic injury, the 29D7 TrkB agonist antibody enhances neuronal survival and promotes functional recovery [31,32,33,34].

There is considerable interest in the possibility of utilizing the receptor-mediated transcytosis pathways that exist on brain endothelial cells that form the BBB to carry biotherapeutics from the blood to the brain parenchyma with the transferrin receptor 1 (TfR1) being the most widely studied [35]. TXB4 is a single domain shark variable new antigen receptor (VNAR) antibody specific to TfR1 with enhanced brain penetration which was derived from the previously reported TXB2 VNAR [36] by restricted randomization of the CDR3 domain [37]. We hypothesized that if the TXB4 module was fused to the 29D7 TrkB agonist antibody it would accumulate in the brain following systemic administration to provide neuroprotection following disease or injury. In the present study we produced a bivalent, bispecific TrkB antibody by cloning the variable regions of the 29D7 agonist antibody into human IgG1 and genetically fusing it to the TXB4 brain shuttle.

Our results show that, unlike the unmodified TrkB agonist antibody, the TXB4-TrkB fusion rapidly accumulated in the brain following a single IV injection. We also found that TXB4-TrkB associates with and activates ERK1/2 signaling in TrkB positive cells in the cortex and tyrosine hydroxylase (TH) positive dopaminergic neurons in the substantia nigra compacta (SNc). When tested in the mouse 6-OHDA model of PD, the TXB4-TrkB antibody completely prevented the loss of TH positive neurons throughout the SNc. In conclusion, fusion with the TXB4 module allows the TrkB agonist antibody to reach neuroprotective concentrations in the brain parenchyma following systemic administration, generating a new class of biologic with therapeutic potential in a wide range of neurodegenerative diseases, acute brain injury situations and possibly depression.

## 2. Materials and Methods

**Production of bivalent VNAR-agonist antibody fusions**. The VH and VL domain sequences from the mouse anti-TrkB 29D7 or anti-TrkC agonist antibody 6.4.1 [33,38] were cloned into the constant regions of the human heavy chain IgG1 and human light chain kappa, respectively. The human Fc domain of both antibodies contained the LALA double mutation (Leu234Ala and Leu235Ala) to attenuate effector function [39]. The original TXB2 BBB shuttle [36] was subjected to restricted CDR3 mutagenesis and the TXB4 variant, which showed enhanced brain penetration [37], was used to generate bivalent VNAR-antibody fusions. TXB4 was fused to either the N-terminus of the heavy chain VH domain via a 3xG4S linker (HC2N format) or between the CH1 and CH2 domain via a 3xG4S and a 1xG4S linker (HV2N format), respectively. The TrkC-TXB4 antibody fusion was produced in the HC2N format and used as a control.

All antibodies were expressed in CHO cells by transient transfection. Supernatants were collected and filtered through 0.22 µm membranes and loaded onto HiTrap MabSelect SuRe Protein A columns (GE Healthcare, Chicago, IL, USA) pre-equilibrated with phosphate-buffered saline (PBS, pH 7.4). Antibodies were eluted with 0.1 M glycine, pH 3.5 into neutralizing buffer (1 M Tris-HCl, pH 9.0) and the buffer exchanged to PBS using HiPrep 26/10 Desalting columns (GE Healthcare). Antibody purity was determined by analytical size exclusion chromatography (SEC) using a Superdex200 column.

**Target binding assays**. For ELISAs, Nunc MaxiSorp plates (Thermo Fisher, Waltham, MA, USA) were coated with 100 µL of 1 µg/mL of human TrkB (Sino Biological, 10047H80M, Beijing, China) mouse or human TfR1 (mTfR1 and hTfR1 ectodomains produced internally) and incubated at 4 °C overnight. Plates were incubated with a blocking buffer (2.5% non-fat dry milk in PBS with 0.1% Tween20, PBST) for 1 h at RT. Purified proteins were mixed with non-fat dry milk in PBST to a final concentration of 2.5% and incubated for 30 min. The blocked protein solutions (100 µL) were transferred to the blocked plates and incubated for 1 h. The plates were washed with PBST and incubated with a goat anti-human Fc−peroxidase antibody diluted 1:5000 (Sigma) in blocking buffer for 30 min. The plates were washed and developed with SureBlue (VWR). The reaction was measured at 370 nm in real time for Vmax analysis and EC50 values were calculated using 4-parametric non-linear regression (Prism). Receptor binding kinetics were measured by surface plasmon resonance (SPR) using a Biacore T200 (GE Healthcare) as described [36]. In brief, a Fc-capture kit (GE Healthcare) was used to immobilize ligand in 0.1% BSA in HBS-EP+ buffer (GE Healthcare). Analyte binding was measured using the single cycle kinetic SPR method in HBS-EP+ at a flow rate 30 µL/min. A flow cell without ligand captured served as a reference. Sensorgrams were fitted using a 1:1 binding model and kinetic constants were determined using Biacore T200 Evaluation software.

**Trk receptor reporter assay**. TrkB- and TrkC-β-lactamase reporter cell lines (CellSensor NFAT-bla CHO-K1, Invitrogen) were passaged twice weekly in DMEM-GlutaMAX medium (Gibco) supplemented with 10% dialyzed fetal bovine serum (dFBS), 100 U/mL penicillin, 100 µg/mL streptomycin, 5 µg/mL blasticidin, 200 µg/mL zeocin, 0.1 mM non-essential amino acid solution (NEAA), and 25 mM HEPES buffer (all from Sigma). For the assay, 2 × 10^4^ cells were seeded per well of black-wall clear-bottom 96-well plates (Corning, Corning, NY, USA) in 100 mL of the same medium but with 0.5% dFBS and incubated overnight at 37 °C in 5% CO_2_. Antibodies were diluted in assay media and 50 µL was added per well to achieve a final concentration range as indicated in the results. Cells were incubated with the antibodies for 4 h at 37 °C, 5% CO_2_ before the addition of 30 µL of the fluorescence resonance energy transfer (FRET) substrate CCF2-AM (ThermoFisher, Waltham, MA, USA). After incubation at room temperature for 90 min protected from light, conversion to CCF2 was measured by a shift in FRET emission. The excitation filter was set at 405 nm, and the emission filters at 460 and 530 nm (FlexStation, Molecular Devices, San Jose, CA, USA) and the ratio of the emission wavelengths (λ1/λ2) was calculated as a measure of β-lactamase activity driven by TrkB or TrkC receptor activation.

**Animal studies**. All in vivo studies were performed in accordance with UK Animals Scientific Procedures Act (1986) and were approved by King’s College London Animal Welfare and Ethical Review Body. A total of 65 adult BalbC mice (8–12 weeks old, Envigo) were used: 45 for the brain accumulation study, three for the brain localization studies and 17 for the 6-OHDA neuroprotection study. All animals were maintained on a 12:12 h light/dark cycle with food and water available *ad libitum*.

**Brain accumulation of injected antibodies by ELISA**. Female BalbC mice were injected IV with molar equivalent doses of either unmodified antibodies (3.6 mg/kg = 25 nmoL/kg) or bivalent TXB4-antibody fusions (4.3 mg/kg = 25 nmoL/kg). Animals were euthanized at 30 min, 1, 2, 4 or 18 h post injection by phenobarbital overdose (1 mL of 200 mg/mL Euthatal) before intracardiac perfusion with PBS. Brains were dissected into left and right hemispheres and stored at −80 °C. Tissue samples were homogenized in 3:1 (*v*/*w*) of PBS containing 1% Triton X-100 supplemented with protease inhibitors (cOmplete^TM^, Sigma) using the TissueRuptor (Qiagen, Hilden, Germany) at medium speed for 10 s and then incubated for 30 min on ice. Lysates were centrifuged at 17,000× *g* for 20 min, and the supernatant was blocked overnight at 4 °C in 2.5% milk in PBS with 0.1% Tween 20. MaxiSorp plates (ThermoFisher) were coated with 100 µL of goat anti-human Fc antibody (Sigma) diluted 1:500 in PBS overnight at 4 °C. The plates were washed and incubated with blocking buffer for 1 h at room temperature. Blocked brain lysates (100 µL) were added to the blocked plates and incubated for 1 h at room temperature. After washing, plates were incubated with goat anti-human Fc-HRP conjugated antibody (Sigma) diluted 1:5000 for 1 h. Plates were then washed and developed with tetramethylbenzidine and the reaction was stopped with 1% HCl. Absorbance was measured at 450 nm and antibody concentrations were determined using standard curves prepared separately for each antibody.

**Brain localization of injected antibodies by immunohistochemistry**. All antibodies were injected at 10 mg/kg SC into male BalbC mice. The animals were euthanized after 18 h by phenobarbital overdose before being intracardially perfused with PBS followed by 10% neutral buffered formalin (Sigma, St. Louis, MO, USA). Brains were removed and submerged in 10% neutral buffered formalin for 24 h and embedded in paraffin wax. Serial 7 µm sagittal sections were dewaxed (2 × 5 min in xylene, 4 × 2 min 100% IMS) and endogenous peroxidases quenched by immersion in 3% H_2_O_2_ for 10 min. Antigen retrieval was performed by boiling sections in 1 mM citric acid at a pH of 6.0 for 10 min. A blocking solution containing 3% porcine serum albumin in 0.05 M tris buffered saline (TBS, pH 7.6) was applied for 90 min before sections were incubated with primary antibodies at 4 °C overnight in a humidified chamber. IV-injected antibodies were detected in the brain with a biotinylated goat anti-human IgG (Vector Laboratories, BA-3000, diluted 1:500, Newark, CA, USA). Marker proteins were detected with chicken anti-TH (Abcam ab76442, diluted 1:1500, Cambridge, UK), rabbit anti-pErk1/2 (Cell Signaling Technology 9101, diluted 1:250, Danvers, MA, USA), and rabbit anti-TrkB (Abcam ab18987). Sections were washed for 2 × 5 min in 0.025% Triton-X100 in TBS before incubation with fluorescent-tagged secondaries (all from ThermoFisher, diluted 1:500, Waltham, MA, USA) for 90 min at room temperature. Biotinylated anti-human IgG was detected with streptavidin-Alexa Fluor 647 and the following were used to detect marker proteins: goat anti-chicken-Alexa Fluor 488; donkey anti rabbit-Alexa Fluor 488; and donkey anti-rabbit Alexa Fluor 647. After secondary incubations, sections were washed for 2 × 5 min in 0.025% Triton-X100 in TBS before incubation with 0.1% Sudan Black B in 70% ethanol for 20 min at room temperature to quench autofluorescence. Slides were washed under running water, dried, and coverslips mounted in Vectashield with DAPI (Vector Laboratories, H-1800). Fluorescence images were acquired using a Zeiss 710 confocal microscope and Axiovision image analysis software.

**6-OHDA unilateral lesion model of Parkinson’s disease**. A single SC injection of either 5 mg/kg of the TrkB antibody, TXB4-TrkB antibody fusion or TXB4-TrkC fusion or 5 mL/kg of PBS was given to male BalbC mice 24 h prior to lesioning. A second 2.5 mg/kg dose of each antibody or PBS was administered at post-lesion day seven. For 6-OHDA lesioning, anesthesia was induced with 5% isoflurane/oxygen and animals were placed in a stereotaxic frame with blunt ear bars. Anesthesia was maintained at 3% isoflurane/oxygen and body temperature was maintained at 37 °C. The surgical site was sterilized with 0.4% chlorhexidine before making an antero-proximal incision along the scalp. Fine-bore holes (Ø 0.5 mm) were made in the skull at coordinates AP: +0.5 mm and ML: +2.2 mm (relative to bregma and skull surface) through which a blunt-ended 30-gauge needle was inserted to DV: −3.5 mm. 6-OHDA.HBr (4 μg in 3 μL 0.02% ascorbate/saline) was infused unilaterally into the striatum (0.5 μL/min) and the needle withdrawn 5 min later. This dose was predicted to produce a partial lesion over a two-week period [40]. Animals received a single dose of buprenorphine (Vetergesic; 0.1 mg/kg, SC) after suturing and 1 mL of rehydrating Hartmann’s solution was administered SC daily for 5 days. One animal in the TrkB antibody group failed to recover adequately from surgery and was excluded from the study.

**Immunohistochemical assessment of TH+ cell bodies in the SNc**. On post-lesion day 14, animals were euthanized by phenobarbital overdose before intracardiac perfusion with PBS followed by 10% neutral buffered formalin (Sigma). Brains were removed and submerged in 10% neutral buffered formalin for 24 h before being embedded in paraffin wax. Serial 7 µm coronal sections encompassing the rostral, medial, and caudal SNc were obtained and processed for TH staining. Sections were dewaxed (2 × 5 min in xylene, 4 × 2 min in industrial methylated spirits) and endogenous peroxidases quenched by immersion in 3% H_2_O_2_ for 10 min. Antigen was retrieved by boiling sections in 1 mM citric acid with a pH of 6.0 for 10 min. Blocking solution containing 1% bovine serum albumin in TBS was applied for 10 min before sections were incubated with primary polyclonal rabbit anti-TH antibody (Millipore ab152, diluted 1:500) at room temperate overnight in a humidified chamber. Sections were washed for 5 min in TBS before incubation with biotinylated goat anti-rabbit secondary antibody (Vector Laboratories BA1000, diluted 1:500) at room temperature for 1 h. Sections were washed for 5 min in TBS before detection with Vectastain Elite ABC Kit (Vector Laboratories, PK6100) followed by the DAB substrate Kit (Vector Laboratories, SK4100). Sections were rinsed in distilled H_2_O for 10 min, dehydrated in 100% IMS (4 × 2 min), cleared in xylene (2 × 5 min) and then mounted with coverslips using the solvent based plastic DPX (Sigma). Photomicrographs of TH-stained SNc sections (3–6 sections per mouse at each of the caudal [AP: −3.52 mm], medial [AP: −3.16 mm] and rostral [AP: −2.92 mm] levels relative to bregma) were acquired at 20X magnification using a Zeiss Apotome microscope and Axiovision software (Carl Zeiss Ltd., Tokyo, Japan). Image J software was used to manually count viable (intact round cells with a clear nucleus and cytoplasm) TH-positive A9 dopaminergic cells of the SNc in both the lesioned and intact hemispheres. SNc cell number in the lesion hemisphere was calculated as a percentage of that lost in the intact hemisphere (n = 3–5 per group). Data were combined across all three rostro-caudal levels to generate a single average value for each animal and the mean calculated per treatment group.

**Statistical analysis**. All statistical analysis was performed using Graphpad Prism 8 software. The concentration response curve of antibodies in the NFAT reporter assay was analyzed by non-linear regression to give EC_50_ values (n = 3–5 independent experiments per concentration per treatment). Percentage TH+ cell loss in the lesioned relative to the intact SNc in the 6-OHDA PD mouse model was tested for Gaussian distribution by Shapiro-Wilk and parametric statistics applied accordingly. The percentage TH+ cell loss of the lesioned relative to intact SNc was compared across treatment groups by one-way ANOVA followed by Tukey’s HSD, * *p* ≤ 0.05.

## 3. Results

**In vitro activity of TXB4-TrkB agonist antibody fusions**. The TXB4 module was fused to the TrkB antibody in two different formats and binding to TfR1 and TrkB was evaluated (Figure 1A). The TfR1 ELISA binding curve for the HC2N format overlapped with that of the TXB4-human Fc fusion control, whereas the binding curve for the HV2N format was shifted to the right for both species of TfR1 (Figure 1B,C). The calculated ELISA EC50 values for HV2N and HC2N formats were all within a similar range (2.3 nM vs 0.7 nM for mTfR1; 3.2 nM vs 1 nM for hTfR1, respectively). The TrkB ELISA binding curves for the unmodified TrkB antibody and both TXB4-TrkB antibody formats closely overlapped with calculated EC50s of 0.5–0.6 nM (Figure 1D). The binding kinetics to the TrkB receptor of the TXB4-TrkB HV2N format were virtually identical to that of the unmodified TrkB antibody as determined by SPR with KDs of 1.30–1.37 nM (Table 1). However, the dissociation rate was slower for the HC2N format, resulting in a lower KD of 0.4 nM.

The relative agonist activity of the various antibodies was evaluated in the TrkB-NFAT-bla CHO-K1 cell line. While all three antibodies achieved the full agonist activity of BDNF, the dose-response curves were shifted to the right (Figure 1E). The calculated EC50 for the unmodified TrkB agonist antibody was 0.34 nM and was relatively close to 0.11 nM for the BDNF. Fusing the TXB4 module in either format reduced the potency relative to the unmodified antibody. The EC50 for the HC2N format was 6.9 nM (20-fold reduction), while that for the HV2N format was 0.9 nM (3-fold reduction) and this format was selected for further animal studies (Table 2). The control TXB4-TrkC antibody fusion in the HC2N format retained was a full agonist in the TrkC-NFAT-bla CHO-K1 assay and had an EC50 of 3.1 nM (data not shown).

**Brain accumulation of the unmodified TrkB antibody versus the TXB4-TrkB fusion**. Mice were injected IV with 25 nmoL/kg of either the TrkB antibody or TXB4-TrkB fusion (HV2N format), and antibody concentrations in brain and plasma were determined by ELISA at the indicated timepoints (Figure 2A). Brain levels of the TrkB antibody remained low (0.13–0.39 nM) throughout the study, whereas the TXB4-TrkB fusion antibody steadily accumulated over the study period reached 4.7 nM, which represents a 12-fold increase over the unmodified antibody at the 18 h timepoint (Figure 2B). The plasma concentration between the TrkB antibody (218 nM) and the TXB4 fusion (187 nM) were not significantly different (Figure 2C). For the control TXB4-TrkC antibody, the brain concertation averaged 6.5 nM at 18 h after IV in injection with 25 nmol/kg compared to 0.3 nM for the unmodified TrkC agonist antibody.

**Brain Localization of the TXB4-TrkB antibody fusion after peripheral administration**. The unmodified TrkB antibody or TXB4-TrkB antibody fusion (HV2N) was administered by a single SC injection (10 mg/kg). Perfused brains were harvested 18 h later and stained by immunohistochemistry using an anti-human IgG to detect the injected antibodies. At lower magnification, no TrkB agonist antibody staining was seen anywhere in the brain (Figure 3A). In stark contrast, the TXB4-TrkB antibody fusion was readily detected within endothelial cells that line capillaries throughout the brain (Figure 3B). If the TrkB or TXB4-TrkB antibodies engage TrkB receptors on the cell surface, internalization into those cells would be expected [41]. The most conspicuous staining for the TrkB receptor was the intracellular staining of large neurons in the cortex (inserts in Figure 3C,D). The TXB4-TrkB antibody fusion was also readily seen in these cells (Figure 3C), but the unmodified TrkB antibody was not detected (Figure 3D). Activation of the ERK1/2 signaling pathway is a canonical feature of neurotrophin receptor activation, which can be detected by antibodies that recognize the phosphorylated form of ERK1/2 [2]. Activated ERK1/2 was not detected by this method in the brains of animals treated with the TrkB agonist antibody (Figure 3E); however, it was readily detected in the brain following treatment with TXB4-TrkB antibody fusion and this was most obvious in the cortex (Figure 3F).

**Target engagement in the 6-OHDA model of PD**. When delivered locally, BDNF limits the death of TH+ dopaminergic neurons in the SNc that normally accompanies the injection of 6-OHDA into the striatum [42,43,44,45]; to this end we have tested if the systemic administration of TXB4-TrkB is neuroprotective in this model of PD. We first wanted to determine if the systemically administered TXB4-TrkB antibody fusion localizes to the SNc in a healthy animal. Using tissue from the brain localization studies in Figure 3, the presence of the TrkB agonist antibody was not within the vicinity of TH+ cells in the SNc 18 h after SC delivery as expected (Figure 4A). In contrast, TXB4-TrkB was readily seen in this brain region 18 h after SC administration (Figure 4B). Moreover, while ERK1/2 is not activated in TH+ neurons following TrkB agonist antibody administration (Figure 4C), it is clearly activated in TH+ neurons following TXB4-TrkB administration (Figure 4D). In contrast, the TXB4-TrkC antibody fusion was not detected in the SNc at the same dose and timepoint, nor was pERK1/2 detected in TH+ cells (data not shown).

**Neuroprotection in the 6-OHDA model of PD**. To test for neuroprotection, mice were treated with either PBS (control), TrkB agonist antibody, the TXB4-TrkC or the TXB4-TrkB antibody fusions (5 mg/kg) 24 h before inducing a partial 6-OHDA lesion and again at day seven (with reduced antibody dose of 2.5 mg/kg). Fourteen days after injection of 6-OHDA, brains were isolated and processed for TH immunoreactivity in the rostral, medial, and caudal SNc. As expected, there was a clear reduction (27.30 ± 7.31%) in the number of TH+ cells as a percentage of the intact hemisphere following 6-OHDA treatment in the PBS control (Figure 5). In contrast there was essentially no cell loss in mice treated with the TXB4-TrkB antibody fusion (3.18 ± 2.6%) relative to the PBS control (One-way ANOVA, Tukey’s HSD, * *p* = 0.035). Neuronal loss was still apparent with either the TrkB agonist antibody (12.47 ± 2.97%) or TXB4-TrkC antibody fusion control (20.74 ± 4.11%) treatment and not significantly different from the PBS control (Figure 5).

## 4. Discussion

The BDNF-TrkB signaling pathway is considered a drug target for a wide range of neurological diseases and depression. However, a short half-life of approximately 10 min in plasma [26] and 1 h in CSF [46], and a high isoelectric point (pI~10) that limits its diffusion in tissues [27] have in part hampered clinical development. BDNF can also bind to the p75 neurotrophin receptor (p75NTR) and in some instances activation of this receptor can induce apoptosis [47]. p75NTR is also an integral component of a receptor complex that inhibits axonal growth and the interaction of BDNF with this complex might detract from its regenerative function [48]. The BDNF/p75NTR interaction might also limit the therapeutic potential of BDNF.

TrkB antibodies do not bind p75NTR, in general have a plasma half-life of several days, and are effective if directly delivered to the site of neural injury. The intravitreal delivery of the 29D7 TrkB agonist antibody delays retinal ganglion cell death in models of acute and chronic retinal injury [49,50], whilst intracerebroventricular administration prior to initiation of a neonatal hypoxic-ischemic brain injury in rats significantly increased neuronal survival and behavioral recovery [31]. Intrathecal application of 29D7 improves motor neuron survival and regeneration in models of spinal cord injury and motor neuron degeneration [14]. It follows that there might be opportunities in the development of a version of 29D7 or other TrkB agonist antibodies if they could readily cross the BBB following systemic administration.

Using TfR1 receptor antibodies to deliver cargos across the BBB is not novel [51,52,53]. However, obstacles have been encountered with particular antibodies including the retention of TfR1 antibodies within brain capillaries [54,55], the lysis of TfR1-expressing reticulocytes [56], competition with transferrin for binding to the TfR1 and/or antibody induced targeting of TfR1 to lysosomes [57,58,59,60]. Moreover, the ubiquitous expression of the receptor in peripheral tissues [61] likely contributes to the short plasma half-life reported for TfR1 antibodies [62,63,64]. Various strategies have been employed to overcome these obstacles, including a reduction in binding affinity, valency, and bispecific antibody formatting. Interestingly, high-affinity single domain VNAR antibodies to TfR1 do not share these problems. The small VNAR domain (12–15 kDa) with an extended CDR3 loop can engage cryptic epitopes inaccessible to standard immunoglobulins [65], which may enable the selective tissue binding, and consequently a longer plasma half-life observed with TXB2 shuttle [66]. TXB4 was derived from TXB2 shuttle by CDR3 mutagenesis and was selected for these studies for its improved physiochemical properties and enhanced brain penetration [37,67].

A brain penetrant TrkB agonist antibody was configured with the TXB4 shuttle in two different formats. In one version, the VNAR domain was fused to the N-terminus of the HC (HC2N), while in the other, it was positioned between the CH1 and CH2 domains above the hinge region (HV2N) to avoid possible steric hinderance of the TrkB paratope. Placing the TXB4 module in the HC2N format had virtually no effect on TfR1 binding and while the ELISA binding curve of the HV2N format was right shifted, the binding EC50 for TfR1 was still in the low nM range. Neither format adversely affected the TrkB binding kinetics of the antibody and while both bispecific formats were full agonists in a TrkB reporter cell assay, the HV2N format was approximately seven-fold more potent. Overall, the N-terminal placement better preserved TfR1 binding, but interference with the TrkB binding paratope resulted in reduced activity. Consequently, the HV2N format was selected for in vivo studies.

After peripheral administration of TXB4-TrkB in the HV2N format (4.3 mg/kg, IV), mouse brain concentrations reached the predicted C_max_ at approximately 18-h post injection [36,68]. The achieved 5 nM brain level was expected to trigger robust activation of endogenous TrkB receptors. Target engagement and activation would be manifested by the uptake of TXB4-TrkB into TrkB expressing cells and the activation of canonical signaling pathways such as the ERK1/2 cascade [69]. Indeed, following a single dose (10 mg/kg, SC) in a healthy animal, we could readily detect TXB4-TrkB within the large TrkB positive neurons in the cortex and also found activation of the ERK1/2 cascade throughout the cortex. Likewise, TXB4-TrkB was found in the SNc and ERK1/2 was clearly activated in the TH+ dopaminergic neurons in the SNc in the same animals. As a control, antibody accumulation and ERK1/2 activation were not seen in the SNc following systemic administration of the unmodified TrkB agonist antibody.

When injected into the striatum, 6-OHDA is taken up into dopaminergic neurons via the high-affinity dopamine transporter. It is then oxidized, and the toxicity of the released reactive oxygen species is reflected in a loss of dopaminergic tracts and terminals in the striatum and cell bodies in the SNc [70]. This is a well-established preclinical model of PD [71], but it is noteworthy that toxicity due to generation of reactive oxygen species might be causative of neuronal loss in a wide range of other neurodegenerative diseases. In this study the unilateral administration of 6-OHDA was associated with the loss of 20%–30% of the neurons throughout the SNc. Remarkably, there was no significant neuronal loss throughout the SNc in animals treated with TXB4-TrkB. Control construct TXB4-TrkC also accumulated in the brain but did not localize to or activate ERK1/2 in the TH+ dopaminergic neurons nor offer any neuroprotection against the 6-OHDA lesion. Future studies will be required to determine the impact of TXB4-TrkB on the development of the motor deficits that are a well-studied trait in this model.

Antibodies can show efficacy in neurodegenerative models when administered without a shuttle due to passive transfer and/or uptake through a disease related or injury induced “leaky” BBB. Indeed, there is evidence suggesting that the BBB is damaged to some extent for a limited period in 6-OHDA models [72]. In this study the unmodified TrkB agonist antibody showed a trend towards limited neuroprotection in the SNc, although it did not reach statistical significance, despite a 3-fold greater potency relative to TXB4-TrkB. While there might have been partial neuroprotection due to some leakage of the TrkB agonist antibody, TXB4-TrkB was by far the more effective treatment. Nonetheless, Han et al., has demonstrated the therapeutic potential of IV administered TrkB agonist antibody Ab4B19 for ischemic brain injury based on leakage across the compromised BBB at the site of injury [73]. While many neurodegenerative diseases are associated with a comprised BBB related to neuroinflammatory processes [74,75], one needs to consider the likely beneficial effects of using a shuttle to deliver a therapeutic antibody to the brain. The most obvious is that the therapeutic antibody can be delivered to regions of the brain where the BBB has yet to be compromised or indeed has recovered from damage. Likewise, the use of a brain shuttle is always likely to result in more antibody reaching the parenchyma, increasing the probability of a therapeutic response. Finally, getting more antibody from the serum to the brain will obviously affect the systemic dose required for a therapeutic effect and minimize the risk of adverse effects.

## 5. Conclusions

In summary, this paper shows that the VNAR TXB4 shuttle targeting the TfR1 can be fused to an agonist antibody to allow for the efficient transport across the BBB and access to the brain parenchyma at physiologically relevant concentrations. The TXB4-TrkB fusion antibody crossed the BBB, accumulated in the brain, and triggered neurotrophin signaling in target cells susceptible to loss in AD (cortical neurons) and PD (dopaminergic neurons in the SNc). Furthermore, systemic treatment with TXB4-TrkB prevented the neuronal loss normally seen in a partial lesion mouse model of PD. As such, TXB4-TrkB can be considered as the first in a new generation of brain penetrant agonist antibodies with therapeutic potential in a wide range of neurodegenerative diseases, acute brain and spinal cord injury, and possibly depression.

## 6. Patents

Stocki, P., Wicher, K.B., Szary, J.M. and Rutkowski, J.L., inventors. Ossianix, Inc., assignee. Improved TfR-selective binding peptides capable of crossing the blood brain barrier International Publication No. WO2019089395A1, published 9 May 2019.

Rutkowski, J.L., Walsh, F., Sinclair, E.H. and Stocki, P., inventors. Ossianix, Inc., assignee. BBB-shuttling VNARs conjugated to neurotrophic agonist antibodies to treat neurodegenerative diseases and conditions. International Publication No. WO2021102276 A1, published 27 May 2021.

## Figures and Tables

**Figure 1 pharmaceutics-14-01335-f001:**
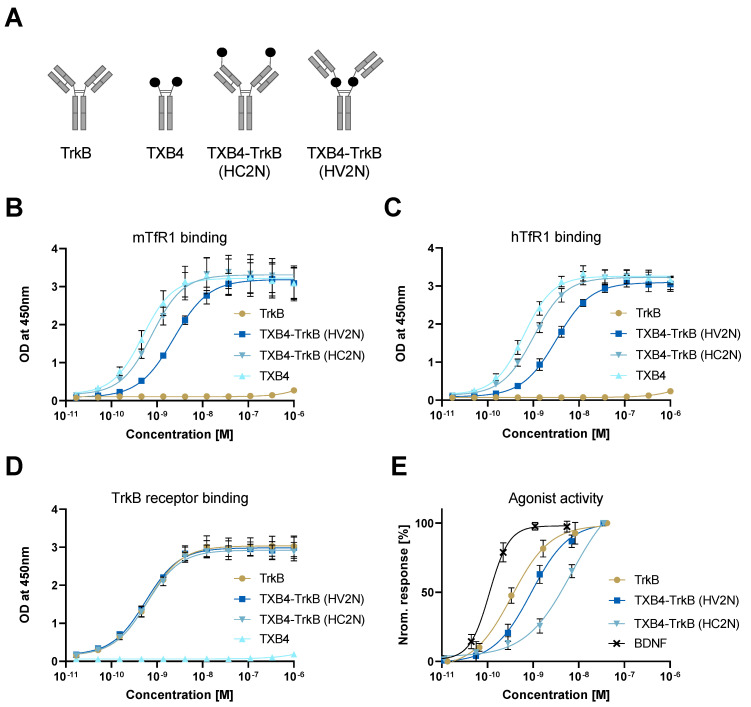
**Binding and agonist activity of TrkB antibody and TXB4—TrkB fusions.** (**A**) Diagram of various antibody formats with the TXB4 VNAR module represented by black-filled circles. The binding of TXB4-TrkB antibody fusion to mTfR1 and (**B**) hTfR1. (**C**) was measured by ELISA compared to the unmodified TXB4 VNAR-hFc and TXB4 module as references. (**D**) Binding to TrkB was measured by ELISA compared to the unmodified TXB4 VNAR-hFc and TXB4 as references compared to the same as references. OD values were used for 4-parametric non-linear regression model to calculate EC50s (±SD, n = 3). (**E**) TrkB and TXB4-TrkB fusion protein were tested for agonist activity using the TrkB-NFAT-bla CHO-K1 reporter cell line assay. Data were normalized and 4-parametric non-linear regression model was used to calculate EC50 values (±SD, n = 3–5) (see Table 2).

**Figure 2 pharmaceutics-14-01335-f002:**
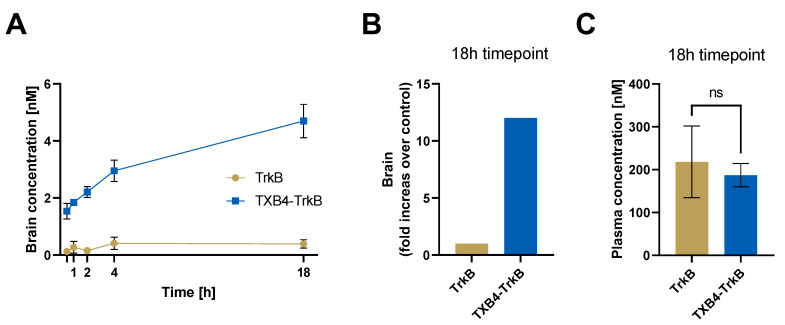
**Brain uptake of TrkB antibody and TXB4-TrkB antibody fusion.** Equimolar doses of the TrkB antibody (3.6 mg/kg) or TXB4-TrkB fusion in the HV2N format (4.3 mg/kg) were administered by single IV injection. Brains were perfused and harvested, and antibody concentrations were determined at the given timepoints. The brain concentration of the TXB4-TrkB fusion rapidly increases after injection (**A**) with brain levels reaching 12-fold over the control by 18 h (**B**), whereas the difference in plasma levels was not significant (ns) as determined by two-tailed, unpaired t-test (**C**). Data are presented as the mean ±SD, n = 3.

**Figure 3 pharmaceutics-14-01335-f003:**
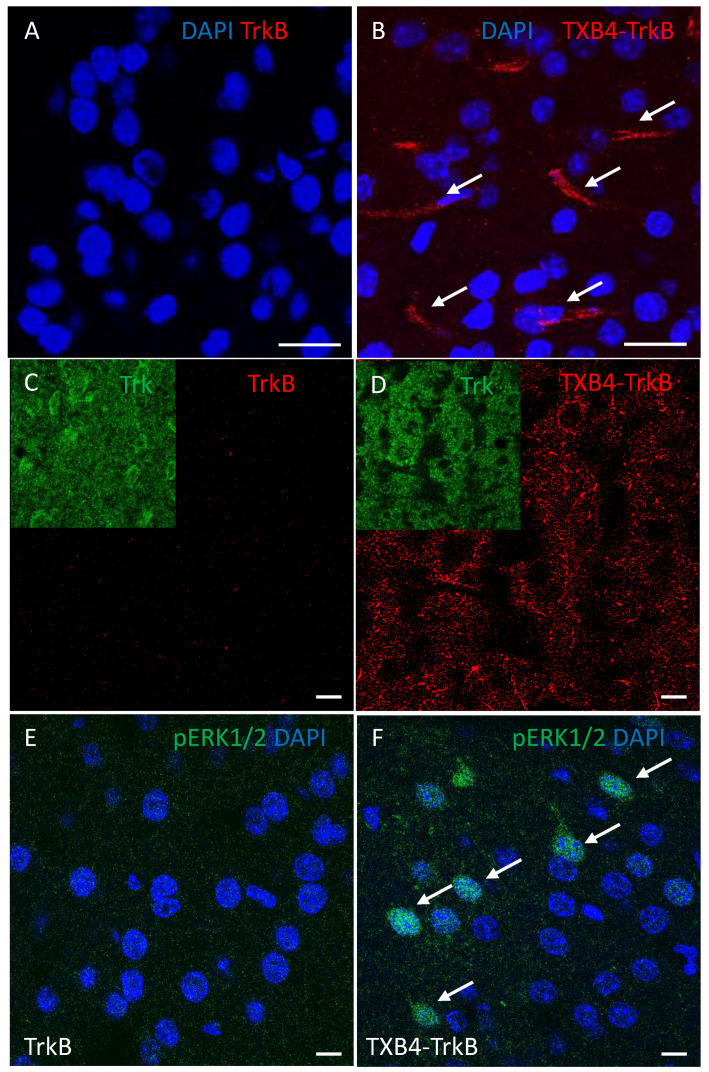
**Brain localization of the TXB4-TrkB antibody fusion with ERK1/2 activation****after peripheral administration.** Perfused brains were harvested 18 h after administration of the TrkB or TXB4-TrkB antibody fusion (10 mg/kg, SC). Human IgG was not detected immunohistochemistry (40×) following TrkB antibody administration (**A**) but was readily seen in endothelial cells (arrows) following TXB4-TrkB administration (**B**). At higher magnification (63×) human IgG was not detected in the cortex following TrkB antibody administration (**C**) but was readily seen in a punctate intracellular pattern in large neurons following TXB4-TrkB administration (**D**). Inserts show the expression of the TrkB receptor in the same field. The same cortical regions as in C and D were also stained with an antibody that recognizes pERK1/2, which was not detected at high magnification (63×) following TrkB agonist antibody administration (**E**) but was readily detected in cells and nuclei (arrows) following TXB4-TrkB administration (**F**). Nuclei were labeled with DAPI in A, B, E and F. Scale bars = 10 μm.

**Figure 4 pharmaceutics-14-01335-f004:**
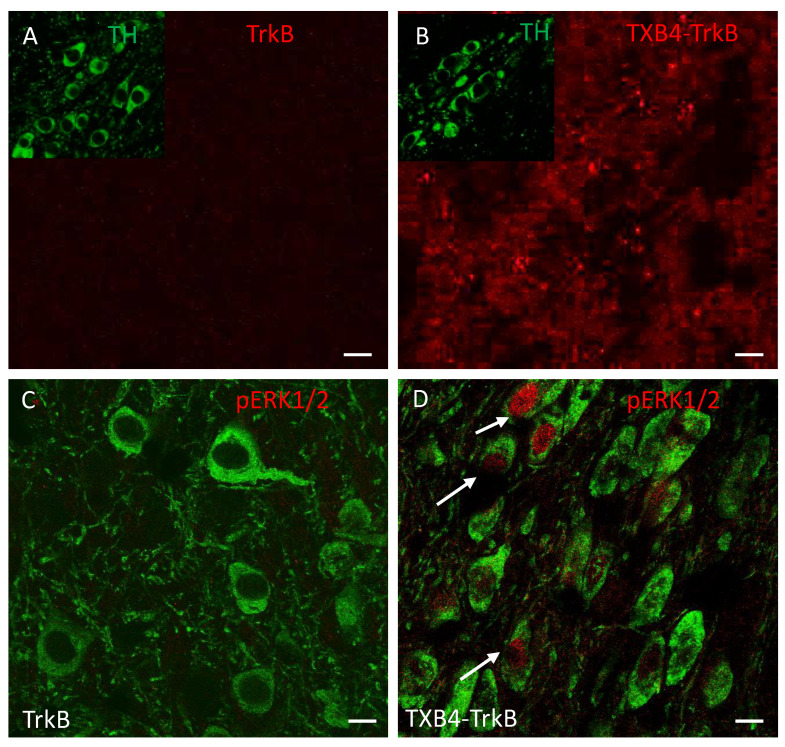
Localization of human IgG, TH and pERK1/2 in the SNc of animals treated with TrkB agonist antibody or TXB4-TrkB. TrkB agonist antibody or TXB4-TrkB were administered SC to mice (10 mg/kg) and brains harvested after 18 h and stained (see methods for details). Within the SNc human IgG (red) was not detected after TrkB agonist antibody administration (**A**) but diffuse and punctate staining was seen following TXB4-TrkB administration (**B**). The inserts in both show dopaminergic neurons within the same field revealed by co-staining for TH. The same regions as in A and B were also co-stained for TH and pERK1/2. pERK1/2 was not detected at high magnification (63×) following TrkB agonist antibody administration (**C**) but was readily detected in some cell nuclei (as indicated by arrows) following TXB4-TrkB administration (**D**). Scale bars = 10 µm.

**Figure 5 pharmaceutics-14-01335-f005:**
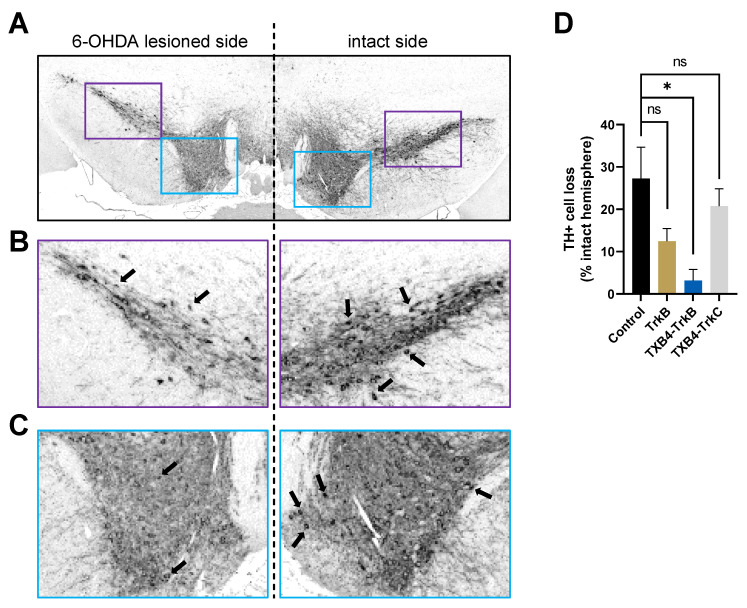
**TXB4-TrkB protects dopaminergic cell bodies in the substantia nigra (SNc) against neuronal loss in the 6-OHDA model of PD.** Representative section of TH+ dopaminergic SNc cells within the 6-OHDA lesioned (left) and intact (right) sides of PBS control (**A**) at 20× magnification. Enlargements of the boxed areas of lateral (**B**) and medial regions (**C**) show examples of the TH+ cell bodies (arrows) counted per section. Cell loss within the 6-OHDA lesioned side relative to the control side is shown (**D**) for groups treated with PBS, TrkB agonist antibody, TXB4-TrkB or TXB4-TrkC fusion antibodies. Data represented as mean ± SEM % TH+ cell loss relative to intact hemisphere ± SEM. One-way ANOVA, Tukey’s HSD, * *p* = 0.035, n = 3–5 mice per group, ns = not significant.

**Table 1 pharmaceutics-14-01335-t001:** Binding kinetics of the TrkB antibody and bivalent TXB4-antibody fusions to the TrkB receptor determined by SPR.

	TrkB Receptor		
	ka 1/[Ms]	kd [1/s]	KD [M]
**TrkB antibody**	2.18 × 10^5^	3 × 10^−4^	1.37 × 10^−9^
**TXB4-TrkB (HV2N)**	2.32 × 10^5^	3.02 × 10^−4^	1.3 × 10^−9^
**TXB4-TrkB (HC2N)**	2.20 × 10^5^	9.77 × 10^−5^	4.44 × 10^−10^

**Table 2 pharmaceutics-14-01335-t002:** Agonist activity using the TrkB-NFAT-bla CHO-K1 reporter cell line assay. A 4-parametric non-linear regression model was used to calculate EC50 values (n = 3–5) (see Figure 1E).

Agonist Activity EC50 [M]				
TrkB Antibody	TXB4-TrkB (HV2N)	TXB4-TrkB (HC2N)	TXB4	BDNF
3.4 × 10^−10^	9.1 × 10^−10^	6.9 × 10^−9^	NA	1.1 × 10^−10^

## Data Availability

Data is contained within the article.

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
