# Peer review of "A Single Domain Shark Antibody Targeting the Transferrin Receptor 1 Delivers a TrkB Agonist Antibody to the Brain and Provides Full Neuroprotection in a Mouse Model of Parkinson’s Disease"

_pharmaceutics, 2022, doi:10.3390/pharmaceutics14071335_

Round 1

Reviewer 1 Report

I really enjoyed reading this manuscript, it was very well written and the results are presented extremely well and the discussion is interesting and comprehensive. The only addition I would ask for is the inclusion of a power analysis for the animal studies. What effect size did you want to see, what power etc, this will explain how you decided on the number of animals to use to yield statistically significant results.

Author Response

Reviewer 1

Comments and Suggestions for Authors

I really enjoyed reading this manuscript, it was very well written and the results are presented extremely well and the discussion is interesting and comprehensive. The only addition I would ask for is the inclusion of a power analysis for the animal studies. What effect size did you want to see, what power etc, this will explain how you decided on the number of animals to use to yield statistically significant results.

The number of animals selected for the experiment was not determined by power analysis. Three to five animals are the standard number commonly used for similar studies in this model. Despite relatively small number of animals, we observed reproducible and statistically significant results, which is indicative that the number of animals in the study was sufficient to produce conclusive results. If the results had not been significant but trending, we would have considered increasing animal number based on power analysis.

Reviewer 2 Report

This article described a series of experiments where a “shuttle-to-the-brain” approach was used to deliver a TrkB agonist antibody. These “shuttle approaches, based on antibodies that bind the transferrin receptor are well known, and not without  caveats. Interestingly, the authors here suggest that single domain shark antibodies that bind to the transferrin receptor 1 (TfR1) on brain endothelial cells have been used to shuttle antibodies and other cargos across the blood brain barrier (BBB) to the brain do not suffer from such deficiencies. Here they use a vNAR named TXB4, which is an improved version of a published vNAR TXB2  (Both TXB2 and TXB4 were previously published, TXB2 in an article and TXB4 in a patent application). For the studies described herein, the TXB4 brain shuttle was fused to a TrkB neurotrophin receptor agonist antibody. They compare it to the un-fused TrkB neurotrophin receptor agonist antibody.

In their results, the authors convincingly show that the TXB4-TrkB fusion retained potent agonist activity at its cognate receptor and after systemic administration showed a 12-fold increase in brain levels over the unmodified antibody. Only the TXB4-TrkB antibody fusion was detected in significant amounts within the brain (followed for 18 hours). Once in the brain, the fusion protein localized to TrkB positive cells in the cortex and tyrosine hydroxylase (TH) positive dopaminergic neurons in the substantia nigra pars compacta (SNc) where it was associated with activated ERK1/2 signaling. When tested in the 6-hydroxydopamine (6-OHDA) mouse model of Parkinson’s disease (PD), the fusion protein TXB4- TrkB, but not the unmodified antibody, completely prevented the 6-OHDA induced death of TH positive neurons in the SNc (presented here as a model for PD). In conclusion, the authors suggest fusion of the TXB4 brain shuttle allows a TrkB agonist antibody to reach neuroprotective concentrations in the brain parenchyma following systemic administration.

The study is well designed, well executed and well presented. The manuscripts is almost flawless, and I have a small number of points I suggest that authors address in what should be considered a “minor revision”.

1) I would like the authors to explain, perhaps in the Discussion. Why they measured the accumulation of antibodies in the for up to 18 hours and not longer. It seems that cMax was not reached at 18 hr (Figure 2A).

2) Along the study the authors used different doses of the antibodies, 25 nmole/kg for measuring accumulation in the brain, 10 mg/kg for efficacy (followed by a 2nd dose at 5 mg/kg). How were these doses chosen, when changing them along the study?

3) TXB4-TrkB was tested in 2 formats, one with the vNAR at an N-terminal position and the second with the vNAR between CH1-CH2. The two behaved differently in antigen binding and in agonist activity, why (I expect some speculation on why in the discussion). The authors present these diffrences in the Results and also in the Discussion (lines 424-432), but do not try to explain why they differ.

4) In Figure 3 and Figure 4 we see images from fluorescent microscopy showing accumulation of TXB4-TrkB in specific addressed in the brain. The absence of “bright field” images showing the corresponding brain structures (like the ones shown in Figure 5) is making the claims associated with these figures less convincing.

Author Response

Reviewer 2

Comments and Suggestions for Authors

This article described a series of experiments where a “shuttle-to-the-brain” approach was used to deliver a TrkB agonist antibody. These “shuttle approaches, based on antibodies that bind the transferrin receptor are well known, and not without caveats. Interestingly, the authors here suggest that single domain shark antibodies that bind to the transferrin receptor 1 (TfR1) on brain endothelial cells have been used to shuttle antibodies and other cargos across the blood brain barrier (BBB) to the brain do not suffer from such deficiencies. Here they use a vNAR named TXB4, which is an improved version of a published vNAR TXB2 (both TXB2 and TXB4 were previously published, TXB2 in an article and TXB4 in a patent application). For the studies described herein, the TXB4 brain shuttle was fused to a TrkB neurotrophin receptor agonist antibody. They compare it to the un-fused TrkB neurotrophin receptor agonist antibody.

In their results, the authors convincingly show that the TXB4-TrkB fusion retained potent agonist activity at its cognate receptor and after systemic administration showed a 12-fold increase in brain levels over the unmodified antibody. Only the TXB4-TrkB antibody fusion was detected in significant amounts within the brain (followed for 18 hours). Once in the brain, the fusion protein localized to TrkB positive cells in the cortex and tyrosine hydroxylase (TH) positive dopaminergic neurons in the substantia nigra pars compacta (SNc) where it was associated with activated ERK1/2 signaling. When tested in the 6-hydroxydopamine (6-OHDA) mouse model of Parkinson’s disease (PD), the fusion protein TXB4-TrkB, but not the unmodified antibody, completely prevented the 6-OHDA induced death of TH positive neurons in the SNc (presented here as a model for PD). In conclusion, the authors suggest fusion of the TXB4 brain shuttle allows a TrkB agonist antibody to reach neuroprotective concentrations in the brain parenchyma following systemic administration.

The study is well designed, well executed and well presented. The manuscript is almost flawless, and I have a small number of points I suggest that authors address in what should be considered a “minor revision”.

1. I would like the authors to explain, perhaps in the Discussion. Why they measured the accumulation of antibodies in the for up to 18 hours and not longer. It seems that cMax was not reached at 18 hr (Figure 2A).

We agree that the presented figure 2A leaves the impression that Cmax was not reached. However, our previous published work with the TXB2 shuttle as well as unpublished results with TXB4 showed in fact that the Cmax was expected at approximately at 1 day post injection (Sehlin, Stocki et al. 2020, Stocki, Szary et al. 2021). Also, the experiment was design mainly to confirm brain penetration of the antibody upon fusion to the brain shuttle. More detailed experiments would be required to obtain precise pharmacokinetic parameters of the fusion molecule, which goes beyond the scope of this manuscript.

Changes to the discussion were included to explain the timing – (lines 438-440).

2. Along the study the authors used different doses of the antibodies, 25 nmole/kg for measuring accumulation in the brain, 10 mg/kg for efficacy (followed by a 2nd dose at 5 mg/kg). How were these doses chosen, when changing them along the study?

We used 10mg/kg, followed by 5mg/kg as uptake studies would predict maximal target engagement at or around these doses. However, future studies will be required to determine optimal dosing regimen.

3. TXB4-TrkB was tested in 2 formats, one with the vNAR at an N-terminal position and the second with the vNAR between CH1-CH2. The two behaved differently in antigen binding and in agonist activity, why (I expect some speculation on why in the discussion). The authors present these differences in the Results and also in the Discussion (lines 424-432), but do not try to explain why they differ.

Such changes in affinity and activity are quite common with bi-specific antibodies. Usually there is a reduction in binding and/or activity of bi-specific constructs in comparison to the parental mono-specific antibodies. These changes mainly result from steric hindrance and accessibility of paratopes to engage with target antigens. Therefore, often multiple formats are tested before selecting the one with the least compromised activity. In this case HV2N showed 7-fold higher activity over HC2N and was selected for further assessment.

Changes to the discussion were introduced to explain it in more detail (lines 426-437).

4. In Figure 3 and Figure 4 we see images from fluorescent microscopy showing accumulation of TXB4-TrkB in specific addressed in the brain. The absence of “bright field” images showing the corresponding brain structures (like the ones shown in Figure 5) is making the claims associated with these figures less convincing.

Figures 3 and 4 low magnification bright field images of brain structure are apparent and, in our view, didn’t add value to the interpretation of the experiment. Therefore, in this circumstance we decided not to include it. 

Sehlin, D., P. Stocki, T. Gustavsson, G. Hultqvist, F. S. Walsh, J. L. Rutkowski and S. Syvanen (2020). "Brain delivery of biologics using a cross-species reactive transferrin receptor 1 VNAR shuttle." FASEB J.

Stocki, P., J. Szary, C. L. M. Rasmussen, M. Demydchuk, L. Northall, D. B. Logan, A. Gauhar, L. Thei, T. Moos, F. S. Walsh and J. L. Rutkowski (2021). "Blood-brain barrier transport using a high affinity, brain-selective VNAR antibody targeting transferrin receptor 1." FASEB J 35(2): e21172.